# Asgard archaea modulate potential methanogenesis substrates in wetland soil

Luis E. Valentin-Alvarado [1,2,14], Kathryn E. Appler [3,14], Valerie De Anda[3,4], Marie C. Schoelmerich[1,12], Jacob West-Roberts[5], Veronika Kivenson[1], Alexander Crits-Christoph[1,2,13], Lynn Ly [6], Rohan Sachdeva [1], Chris Greening [7,8], David F. Savage [1,9,10], Brett J. Baker [3,4] ✉ & Jillian F. Banfield [1,5,7,11] ✉

The roles of Asgard archaea in eukaryogenesis and marine biogeochemical cycles are well studied, yet their contributions in soil ecosystems remain unknown. Of particular interest are Asgard archaeal contributions to methane cycling in wetland soils. To investigate this, we reconstructed two complete genomes for soil-associated Atabeyarchaeia, a new Asgard lineage, and a complete genome of Freyarchaeia, and predicted their metabolism in situ. Metatranscriptomics reveals expression of genes for [NiFe]-hydrogenases, pyruvate oxidation and carbon fixation via the Wood-Ljungdahl pathway. Also expressed are genes encoding enzymes for amino acid metabolism, anaerobic aldehyde oxidation, hydrogen peroxide detoxification and carbohydrate breakdown to acetate and formate. Overall, soil-associated Asgard archaea are predicted to include non-methanogenic acetogens, highlighting their potential role in carbon cycling in terrestrial environments.

Wetland soils are hotspots for methane production by methanogenic archaea. The extent of methane production depends in part on the availability of substrates for methanogenesis (e.g., hydrogen, carbon dioxide, formate, formaldehyde, methanol, acetate), compounds that are both produced and consumed by co-existing microbial community members. Among the groups of organisms that coexist with methanogens are Asgard archaea, of recent interest from the perspective of eukaryogenesis[1–4]. To date, numerous lineages of Asgard archaea have been reported from anaerobic, sedimentary freshwater, marine, and hydrothermal environments[1–15]. Predictions primarily from draft metagenome-assembled genomes (MAGs) indicate metabolic diversity and flexibility that may enable them to occupy these diverse ecological niches. It appears that Asgard archaea are not capable of methane production since they lack the key canonical methyl-coenzyme M reductase (MCR). Although a few complete genomes for Asgard from hydrothermal and geothermal environments have been reported[9,15–17], most metabolic analyses of Asgard archaea are limited by reliance on partial genomes. To date, no Asgard genomes from non-estuarine wetland soils have been reported. Thus, nothing is known about the ways in which Asgard archaea directly (via methane production) or indirectly (via metabolic interactions) impact methane cycling in wetlands.

[1]Innovative Genomics Institute, University of California, Berkeley, California, USA. [2]Department of Plant and Microbial Biology, University of California, Berkeley, CA, USA. [3]Department of Marine Science, University of Texas at Austin; Marine Science Institute, Port Aransas, TX, USA. [4]Department of Integrative Biology, University of Texas at Austin, Austin, TX, USA. [5]Environmental Science, Policy and Management, University of California, Berkeley, CA, USA. [6]Oxford Nanopore Technologies Inc, New York, NY, USA. [7]Department of Microbiology, Biomedicine Discovery Institute; Monash University, Clayton, VIC, Australia. [8]Securing Antarctica's Environmental Future, Monash University, Clayton, VIC, Australia. [9]Howard Hughes Medical Institute, University of California, Berkeley, California, USA. [10]Department of Molecular and Cell Biology, University of California Berkeley, Berkeley, USA. [11]Earth and Planetary Science, University of California, Berkeley, CA, USA. [12]Present address: Department of Environmental Systems Sciences; ETH Zürich, Zürich, Switzerland. [13]Present address: Cultivarium, Watertown, MA, USA. [14]These authors contributed equally: Luis E. Valentin-Alvarado, Kathryn E. Appler. ✉e-mail: acidophile@gmail.com; jbanfield@berkeley.edu

To investigate the roles of Asgard archaea in carbon cycling in wetland soil, we reconstructed two complete genomes for a newly defined group, here named Atabeyarchaeia, and one complete genome for a group named Freyarchaeia. Freyarchaeia MAGs were originally reconstructed from Guaymas Basin, located in the Gulf of California, México[14], and from Jinze Hot Spring (Yunnan, China)[4]. Subsequently, another group used the original data to recover similar genomes and referred to them as Jordarchaeia[18]. Here, we retain the original nomenclature. The genomes for soil Asgard archaea were initially reconstructed by manual curation of Illumina short-read assemblies and then validated using both Nanopore and PacBio long reads. These fully curated genomes enabled us to comprehensively sample the genetic potential. This enabled us to constrain metabolism without the risks associated with reliance on partial and/or contaminated draft genomes. The genomes also provided context for metatranscriptomic measurements of in situ activity. Our integrated analysis of gene expression and metabolic predictions revealed roles for Atabeyarchaeia and Freyarchaeia in the production and consumption of carbon compounds that can serve as substrates for methanogenesis by coexisting methanogenic archaea.

## Results

### Complete genomes and phylogenetic placement of Asgard archaea from wetland soil

We analyzed Illumina metagenomic data from samples collected from 20 cm to 175 cm depth in the soil of a wetland located in Lake County, California, USA. We previously reported megaphages[19] and *Methanoperedens* archaea and their 1 Mb-scale "Borg" extrachromosomal elements from this site[20] (Fig. 1a). From the metagenomic analyses conducted at this site, we determined that archaea account for >45% of the total community below a depth of 60 cm. Archaeal groups detected include members of the Asgardarchaeota, Thermoproteota, Thermoplasmatota, Nanoarchaeota, Micrarchaeota, Halobacteriota, Hadarchaeota, Aenigmatarchaeota and Methanobacteriota (Fig. 1b). Notably, methanogens assigned to Methanomicrobia and Methanomethylicia were actively transcribing the MCR complex, consistent with methanogenesis and anaerobic oxidation of methane in situ (Supplementary Fig. 1). In addition, we searched publicly available metagenomes to determine the biogeographical distribution of Atabeyarchaeia and Freyarchaeia using singleM[21]. We detected Atabeyarchaeia predominantly in freshwater sediment, peat, and wetland environments somewhat analogous to the site studied here. In

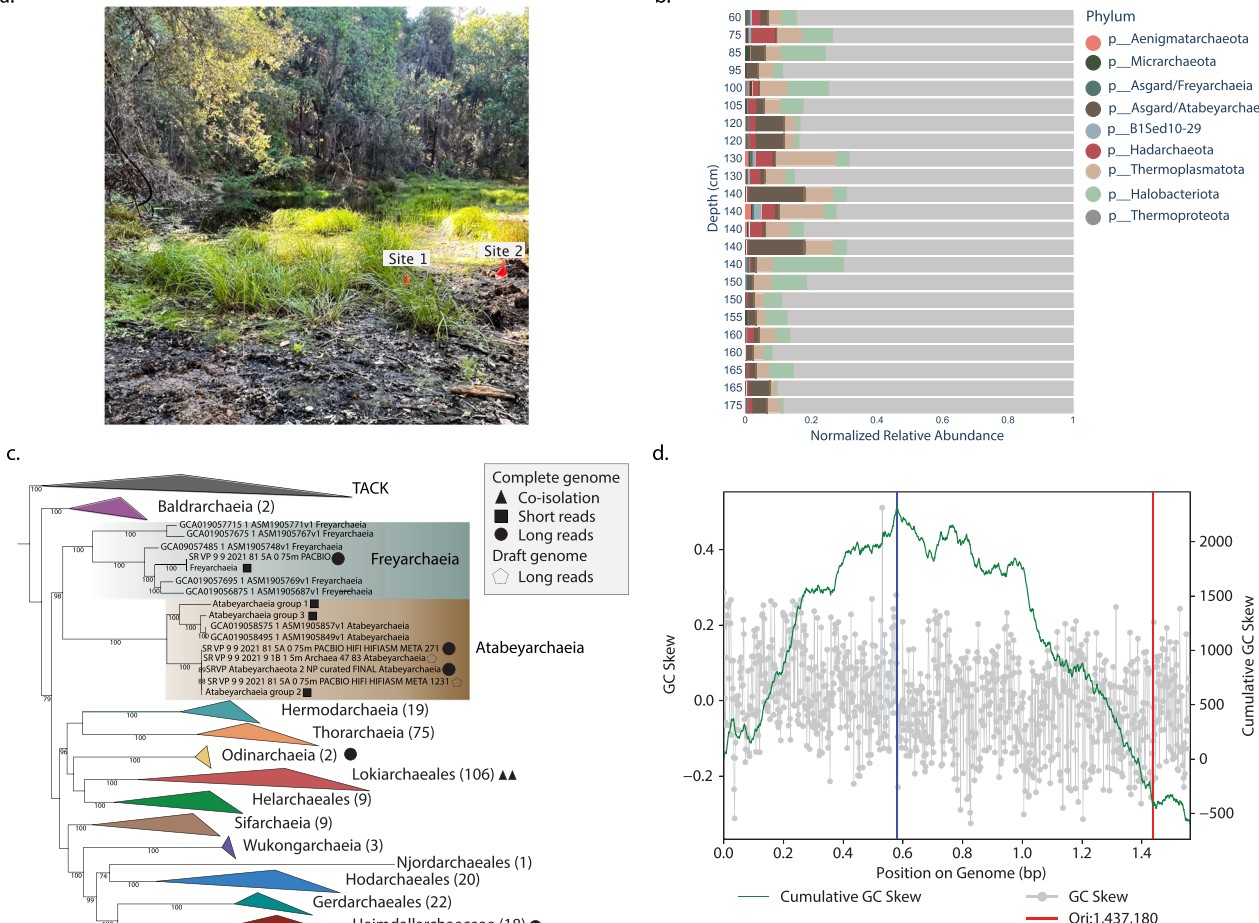

**Fig. 1 | Archaea dominate deep regions of wetland soil and host novel Asgard archaea. a** Photograph of the vernal pool that was sampled for metagenomics analyses in this study, in Lake County, California, USA. **b** Archaeal genomic abundance excluding bacterial genomes. **c** Phylogenetic distribution of Asgard Archaea complete genomes. The maximum-likelihood phylogeny was generated with IQ-TREE v1.6.1, utilizing 47 concatenated archaeal Clusters of Orthologous Groups of proteins (arCOGs). The best-fit model was determined as LG + F + R10 based on the Bayesian Information Criterion. Non-parametric bootstrapping was conducted

with 1000 replicates for robustness. The filled-in square, circle, and triangle indicate closed complete genomes from short reads, published complete genomes from long reads, and genomes from co-isolated cultured representatives, respectively. The Pentagon highlights the long-read draft genomes from this site (PacBio or Nanopore). **d** Indicators of bidirectional replication in Atabeyarchaeia complete genomes. The GC skew is shown as a gray plot overlaying the cumulative GC skew, presented as a green line. The blue lines mark the predicted replication terminus. Source data are provided as a Source Data file.

contrast, Freyarchaeia were found across a diverse array of habitats, including extreme environments such as hydrothermal vents and hot springs, as well as in animal-associated and wetland ecosystems (Supplementary Fig. 2a, b).

From 60 cm, 80 cm, and 100 cm deep wetland soil, we recovered four draft Asgard genomes, three of which were manually curated to completion using methods described previously[22]. Taxonomic classification using the Silva DB placed the 3,576,204 bp genome as Freyarchaeia. 16S rRNA gene sequence analysis showed the two other complete genomes were distinct from Freyarchaeia (16S rRNA genes are < 75% identical), thus representing organisms from a separate, new lineage. These genomes are 2,808,651 and 2,756,679 bp in length (Supplementary Data 1) with an average amino acid identity (AAI) of ~70% (Supplementary Data 2).

Phylogenetic analyses using several sets of marker genes (see "materials and methods") placed our two novel complete genomes in a monophyletic group within the Asgard clade as a sister group to Freyarchaeia (Fig. 1c). We performed phylogenetic analyses using concatenated marker sets of 47 archaeal clusters of orthologous genes (arCOGs) and 15 ribosomal protein (RP15) gene cluster (Supplementary Fig. 3), as well as the 16S rRNA gene (Supplementary Fig. 4). The new genomes share only 40–45% AAI when compared to other Asgard genomes, consistent with their assignment to a new phylum. Although our analyses provide evidence for distinction at the phylum level, we chose to adhere to the Genome Taxonomy Database (GTDB) for standardized microbial genome nomenclature (Supplementary Data 3). Here, we propose the name Candidatus "Atabeyarchaeia" for this new group, where 'Atabey' is a goddess in of Taíno Puerto Rican mythology. Atabeyarchaeia is represented by the complete Atabeyarchaeia group 1 (Atabeyarchaeum deiterrae 1 sp. nov.) and group 2 (Atabeyarchaeum deiterrae 2 sp. nov) genomes. Included in this group are two highly fragmented draft genomes from the Lake Cootharaba Group (ALCG)[12]. The cumulative GC skew of the Freyarchaeia and Atabeyarchaeia genomes is consistent with bidirectional replication. This style of replication is typical of bacterial genomes but has not been widely reported in Archaea and has never been described in the Asgard group (Fig. 1d and Supplementary Fig. 5). Intriguingly, complete genome analyses of representatives from Heimdallarchaeia, Lokiarchaeia, and Odinarchaeia suggest the absence of bidirectional replication in these lineages (Supplementary Fig 5).

Unexpectedly, we found that 92% to 95% of tRNA genes from all three genomes contain at least one intron. This contrasts with the general estimate that 15% of archaeal tRNA harbor introns[23], and with Thermoproteales (another order of archaea), where 70% of the tRNAs contain introns[24]. In total, there are 228 tRNA introns across the three new Asgard genomes (Supplementary Data 4). Unlike most archaeal tRNA introns that occur in the anticodon loop at position 37 / 38[25,26], Atabeyarchaeia and Freyarchaeia introns often occur at non-canonical positions, and over half of their tRNA genes have multiple introns (Supplementary Data 4).

Subsequently, we acquired and independently assembled Oxford Nanopore and PacBio long-reads from a subset of the samples to generate three circularized genomes that validate the overall topology of all three curated Illumina read-based genomes (Supplementary Fig. 6 and Supplementary Data 1). These complete genomes allowed us to genomically describe two Atabeyarchaeia-2 strain variants from 100 cm and 175 cm depth soil. In addition, we used Illumina reads to curate a draft Nanopore genome for another Atabeyarchaeia species, Atabeyarchaeia-3, from 75 cm and 175 cm depth soil (Supplementary Fig. 6). The Atabeyarchaeia-3 genome is most closely related to the Asgard Lake Cootharaba Group (ALCG) fragments[18]. To further solidify the phylogenetic position of Atabeyarchaeia, we included the Atabeyarchaeia-3 genome and another draft genome (Atabeyarchaeia-4) from Illumina reads in the phylogenetic analysis.

Using the Asgard clusters of orthologous genes (AsCOGS) database and functional classification, we identified eukaryotic signature proteins (ESPs) in the complete and public genomes of Atabeyarchaeia and Freyarchaeia[2,3]. Atabeyarchaeia and Freyarchaeia genomes had the highest percentage of hits for 'Intracellular trafficking, secretion, and vesicular transport' (U) among the AsCOG functional classes, accounting for 84.3% of the hits to the database. Within this class, we identified key protein domains such as Adaptin, ESCRT-I-III complexes, Gelsolin family protein, Longin domain, Rab-like GTPase, Ras family GTPase, and Roadblock/LC7 domain (Supplementary Data 5 and Supplementary Fig. 8). The 'Post Translational modification, protein turnover, and chaperones' category (O) followed with a count of 101 (15.8%), highlighting domains like Ubiquitin, Jab1/MPN domain-containing protein, and the RING finger domain. The presence of ESPs in the newly described Atabeyarchaeia lineage and their presence in Freyarchaeia aligns with previous findings for Asgardarchaeota[1,3,4].

## Expression of energy conservation pathways constrain key metabolisms in situ

We analyzed the metabolic potential of the three complete genomes and investigated their gene expression in situ through metatranscriptomics of soil samples (see "materials and methods", Fig. 2, Supplementary Data 6 and Supplementary Data 7). The metatranscriptomic data indicate the expression of genes involved in key energy conservation pathways (Fig. 3a). The most transcribed metabolic genes encode soluble heterodisulfide reductase (HdrABC), a cytosolic protein complex involved in electron bifurcation, [NiFe] hydrogenases (groups 3 and 4), ATP synthase, numerous aldehyde ferredoxin oxidoreductases, enzymes for phosphoenolpyruvate (PEP) and pyruvate metabolism, and carbon monoxide dehydrogenase/acetyl CoA synthetase (CODH/ACS). Notably, the Hdr, the group 3 and group 4 hydrogenase (including up to eight Nuo-like), as well as the ATP synthase, are co-encoded in a syntenic block in all the genomes (Fig. 4a).

The group 4 [NiFe]-hydrogenases of these archaea appear to conserve energy through a common mechanism with complex I (Nuo). Phylogenetic analysis of the large subunit of these hydrogenases suggests they are closely related to those of Odinarchaeia, Heimdallarchaeia, and Hermodarchaeia (Fig. 4b and Supplementary Data 8). However, the exact function of this unclassified Asgard group has not been validated biochemically[27]. One clue relies on the identification of eight genes homologous to the transmembrane proton-translocating subunits of complex I, NuoL, M, and N (E. coli nomenclature), and Mrp-type $Na^+/H^+$ antiporters. Thus, these Asgard archaea may couple hydrogenase activity to $Na^+/H^+$ translocation, thereby generating a sodium/proton-motive force that drives the ATP synthase[28–30]. We employed AlphaFold2 to model the hydrogenase and associated complex I-like modules from Freyarchaeia and Atabeyarchaeia (Fig. 4c and Supplementary Fig. 9a–f). Overall, the predicted structure has a cytosolic and membrane-associated portion (Fig. 4c). The cytosolic portion aligned with the respiratory membrane-bound hydrogenase (MBH) from Pyrococcus furiosus[28] with high confidence (Fig. 4d). When superimposed, the predicted structures of the membrane-associated hydrophobic L, M, K, and S chains are equivalent to those of bacterial complex I. In the canonical complex I[31,32], Chains L, M, N, and K translocate protons[32,33], a process that is facilitated by a lateral helix, helix HL, of chain L. A full-length helix HL is also present in the L-like subunit of the Asgard complexes (Fig. 4e), but not in the other structurally characterized respiratory membrane-bound hydrogenases (Fig. 4e and Supplementary Fig. 9a–f).

The Group 3c cofactor-coupled bidirectional [NiFe] hydrogenase (Supplementary Fig. 10a), in combination with HdrABC, suggests the capability to bifurcate electrons from $H_2$ to ferredoxin and an unidentified heterodisulfide compound. This capacity has been observed in methanogenic archaea via the MvhADG–HdrABC system[34,35].

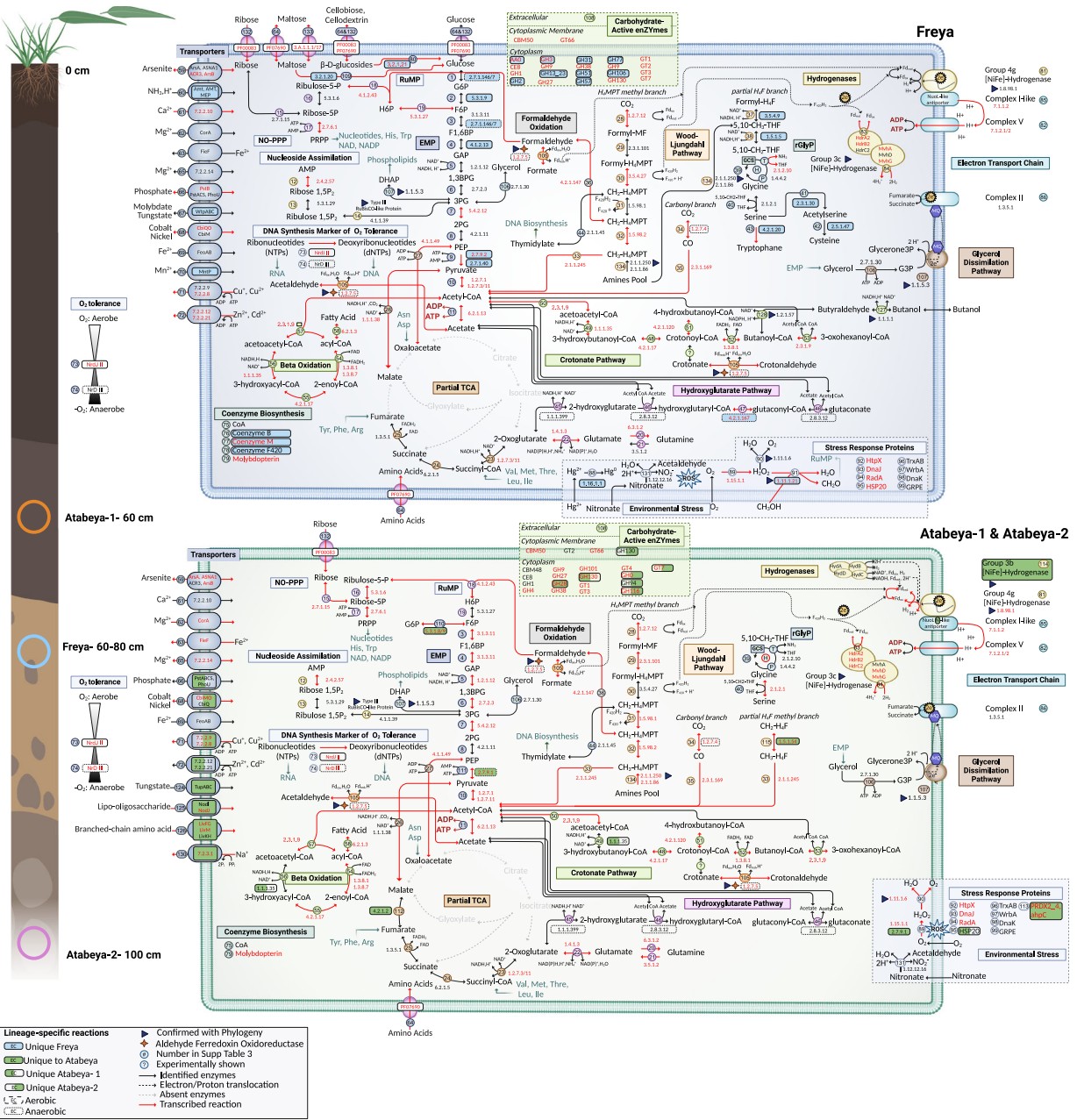

**Fig. 2 | Metabolic capacities of terrestrial Atabeyarchaeia and Freyarchaeia for overall implications for biogeochemical cycling in wetlands.** Inference of the pathways from the complete genomes is based on the comparison of predicted proteins with a variety of functional databases ("see methods"). The extraction depth location within the cores is shown on the left. All reactions are numbers and correspond to Supplementary Data 7. EC/TCDB numbers shaded fully or partially in blue or green are unique to the lineages and complete genomes, whereas the dashed boxes distinguish oxygen-sensitive enzymes. The multi-functional aldehyde ferredoxin oxidoreductase is shown with a star. Proteins marked with a triangle have generated phylogenies to determine their evolutionary histories and substrate specificity. Reactions with mapped transcripts are denoted with red text and arrows. Created using BioRender.com.

Atabeyarchaeia genomes also encode two independent gene clusters for the Group 3b NADP-coupled [NiFe] hydrogenases (Supplementary Fig. 10b). Their presence suggests the capacity to maintain redox equilibrium and potentially grow lithoautotrophically by using $H_2$ as an electron donor, as suggested for other Asgardarchaeota members[10,14,27].

Atabeyarchaeia and Freyarchaeia encode both the tetra-hydromethanopterin ($H_4$MPT) methyl branch and the carbonyl branch of the Wood–Ljungdahl pathway (WLP) (Supplementary Fig. 11). This reversible pathway can be used to reduce $CO_2$ to acetyl coenzyme A (acetyl CoA), which can be further converted to acetate. This last conversion can lead to energy conservation in both Asgard lineages via substrate-level phosphorylation when mediated by acetate-CoA ligase (see below). We confirmed the expression of almost all of the genes of the methyl and carbonyl branches, including the acetate-CoA ligase, in all complete genomes. Gene expression data indicate that when $H_2$ is available in the ecosystem, these archaea have the genetic potential to utilize the Wood-Ljungdahl pathway (WLP) for the reduction of $CO_2$ or formate. In addition, the potential exists for the reverse operation of the WLP to oxidize acetate. The observed expression of energy-converting hydrogenases and ATP synthases suggests that these archaea are equipped to maintain cofactor ratios and conserve energy, which could support these metabolic processes if they occur. The metabolic inferences, along with the transcriptional data, including the

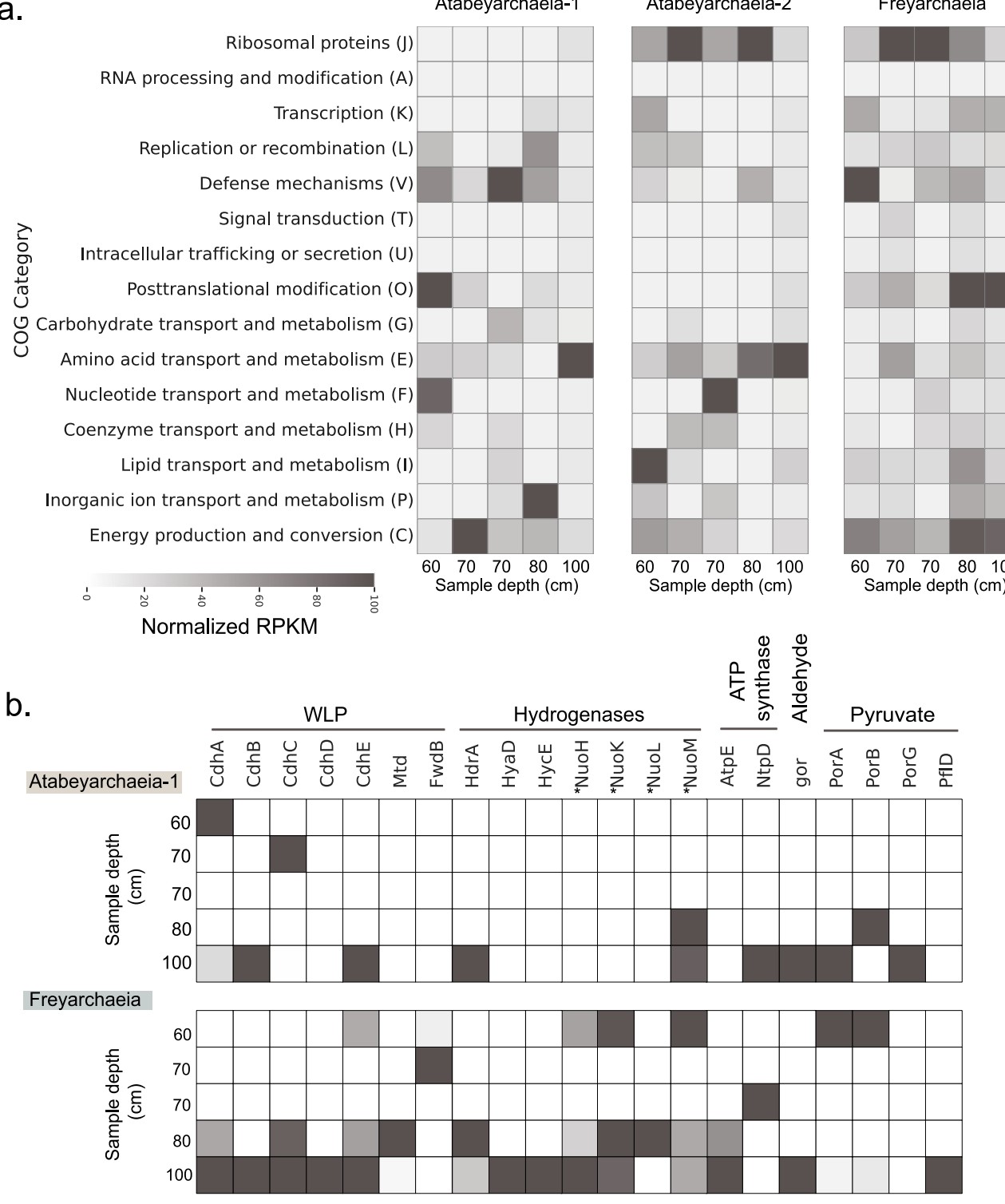

**Fig. 3 | Metatranscriptomic profiling of soil-associated Asgard archaeal genomes. a** Heatmap visualization of normalized Reads Per Kilobase per Million mapped reads (RPKM) values for ORFs with high sequence similarity (≥ 95%) to the genomes of Atabeyarchaeia-1, Atabeyarchaeia-2, and Freyarchaeia, across various soil depths. It is important to note that these samples are homogenized and do not represent biological replicates, thus they do not reflect variation across depths "see methods". A total of 2191 open reading frames (ORFs) were categorized using the Clusters of Orthologous Groups (COG) database, with Atabeya-1, Atabeya-2, and Freya expressing 465, 804, and 922 unique ORFs, respectively. The ORFs were annotated and assigned to 15 COG categories, indicating the functional potential of each archaeal genome in situ. Columns represent metatranscriptomes from different soil depths, highlighting the spatial variability in the expression of key metabolic and cellular processes. **b** Expanded heatmap of Atabeyarchaeia-1 and Freyarchaeia expressed genes under the category C: Energy production and conversion. Key genes of the Wood-Ljungdahl Pathway (CODH/ACS, carbon monoxide dehydrogenase/acetyl-CoA synthase; *fwdB*, formate dehydrogenase; *mtd*, 5,10-methylene-H4-methanopterin dehydrogenase), hydrogenases and associated genes (*hdrA*, heterodisulfide reductase and group 3c NiFe-hydrogenase; *mvh*, methyl viologen reducing hydrogenase); *hyaD* (maturation factor); *hycE* and *nuo*-like subunits denoted with an asterisks, (group 4 NiFe-hydrogenase), ATP synthase (*atpE*, V/A-type H⁺/Na⁺-transporting ATPase subunit K; *ntpD*, V/A-type H⁺/Na⁺ transporting ATPase subunit D) and aldehyde metabolism (*gor*, aldehyde:ferredoxin oxidoreductases), pyruvate oxidation (*porABCD*, 2-pyruvate:ferredoxin oxidoreductase; *pflD*, pyruvate-formate lyase). Source data are provided as a Source Data file.

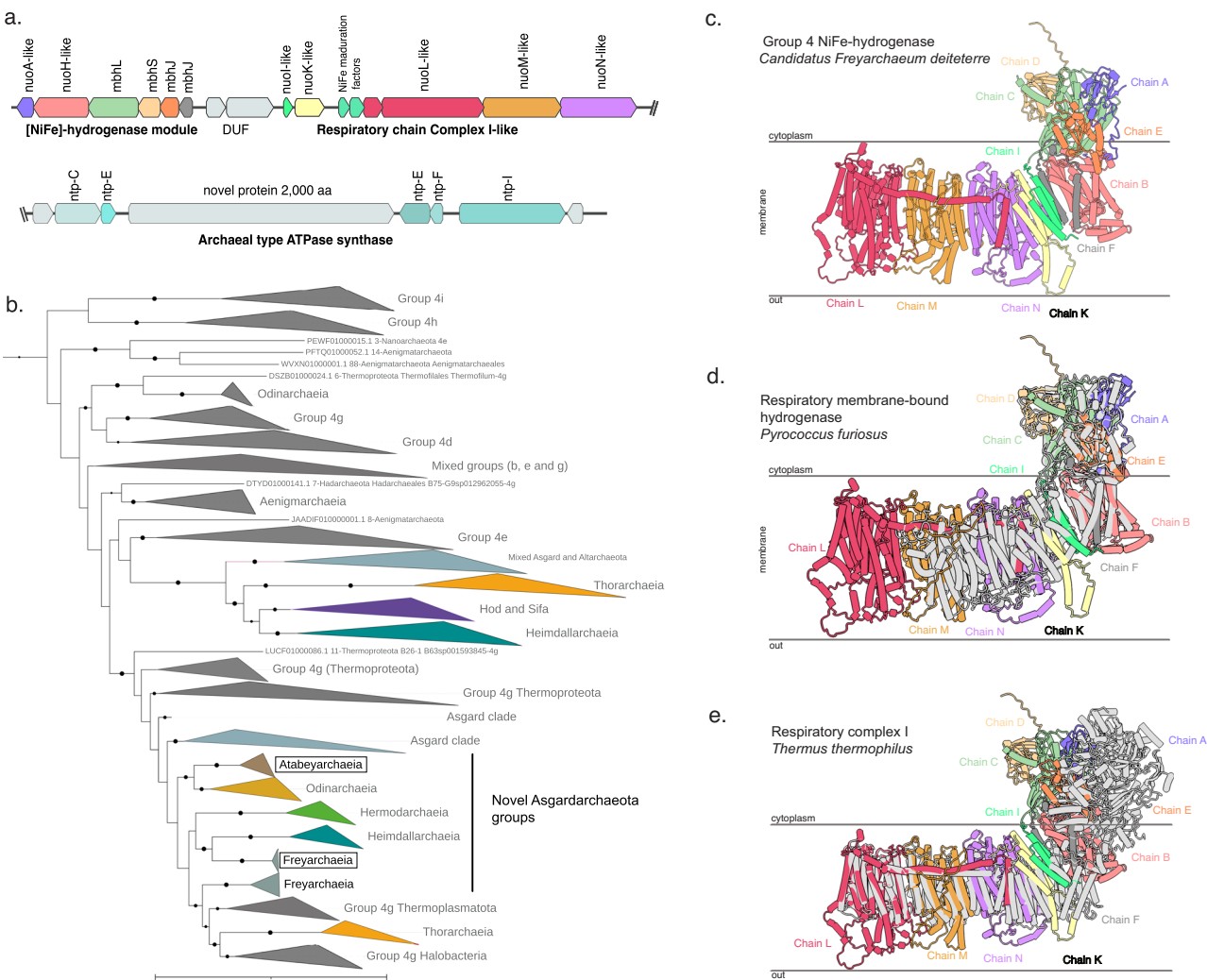

**Fig. 4 | Phylogeny, genetic organization and AlphaFold predicted structure of the novel group 4 energy-conservation complex I-like NiFe-hydrogenase from Asgard archaea. a** Genetic organization of the group 4 [NiFe]-hydrogenase module, the proton-translocating membrane module, and ATP synthase from the Freyarchaeia genome. **b** Maximum likelihood phylogeny of group 4 [NiFe]-hydrogenase large subunit from Asgard archaea and reference sequences. The bolded taxonomic groups highlight the clades with genomes from this study used for modeling. Note that group 4 g [NiFe]-hydrogenase is currently polyphyletic based on HydDB. **c** AlphaFold Multimer models of [NiFe]-hydrogenase module and the proton-translocating membrane module where each candidate subunit is represented by a different color based on the best subunit matched. **d** AlphaFold Multimer model of Freyarchaeia hydrogenase complex colored by chains, aligned with cryoEM structure of a respiratory membrane-bound hydrogenase (MBH) from *Pyrococcus furiosus*[28] (PDB ID: 5L8X). **e** AlphaFold Multimer model of Freyarchaeia hydrogenase complex colored by chains, aligned with the crystal structure of respiratory complex I from *Thermus thermophilus*[32] (PDB: 4HEA).

---

expression of pyruvate-ferredoxin oxidoreductase (por) genes in all three Asgard genomes indicate a reliance on an archaeal version of the WLP to perform acetogenesis[35,36]. This acetogenic lifestyle appears to involve energy conservation through a hydrogenase-dependent chemiosmotic mechanism similar to that observed in some acetogenic bacteria[37].

## Potential for non-methanogenic methylotrophic lifestyle

Despite the absence of the MCR complex, Freyarchaeia genomes have all the necessary genes to synthesize coenzyme-M from sulfopyruvate via the ComABC pathway similar to methanogens[38,39]. Most methanogens conserve energy via the $Na^+$-translocating MtrA-H complex, which is encoded by an eight-gene cluster[40]. Although Atabeyarchaeia and Freyarchaeia do not have the genes for the full complex, Atabeyarchaeia-1 has two copies of a subunit similar to the $CH_3$-$H_4MPT$-dependent methyltransferase subunit A (MtrA-like) and both Freyarchaeia and Atabeyarchaeia also encode the $CH_3$-$H_4MPT$-dependent methyltransferase subunit H (MtrH), along with a phylogenetically distinct fused polypeptide of MtrA-like and MtrH (Fig. 5a). Under the conditions prevalent at the time of sampling, the *mtr* genes were only weakly expressed (< 10 transcripts) (Supplementary Data 6 and Supplementary Data 8). While the biochemical activity of these divergent non-methanogen-associated MtrA-like and MtrH-like enzymes remains unclear, our phylogenetic analyses suggest they are phylogenetically related to methanogenic MtrA, MtrH, and MtrAH sequences. This suggests their potential role in converting $CH_3$-$H_4MPT$ to $H_4MPT$, transferring a methyl group to an acceptor –possibly coenzyme-M, which is predicted to be synthesized by Freyarchaeia. As they lack the MCR complex, the subsequent fate of the methyl group remains uncertain.

Although Atabeyarchaeia and Freyarchaeia genomes do not encode MtrE, we identified genes associated with methyltransferase systems encoded in close proximity to the MtrH gene. Specifically, the genomes are predicted to encode trimethylamine methyltransferase (MttB-like, COG5598 superfamily), undefined corrinoid protein (MtbC-like), and putative glycine cleavage system H (*gcvH*) (Supplementary

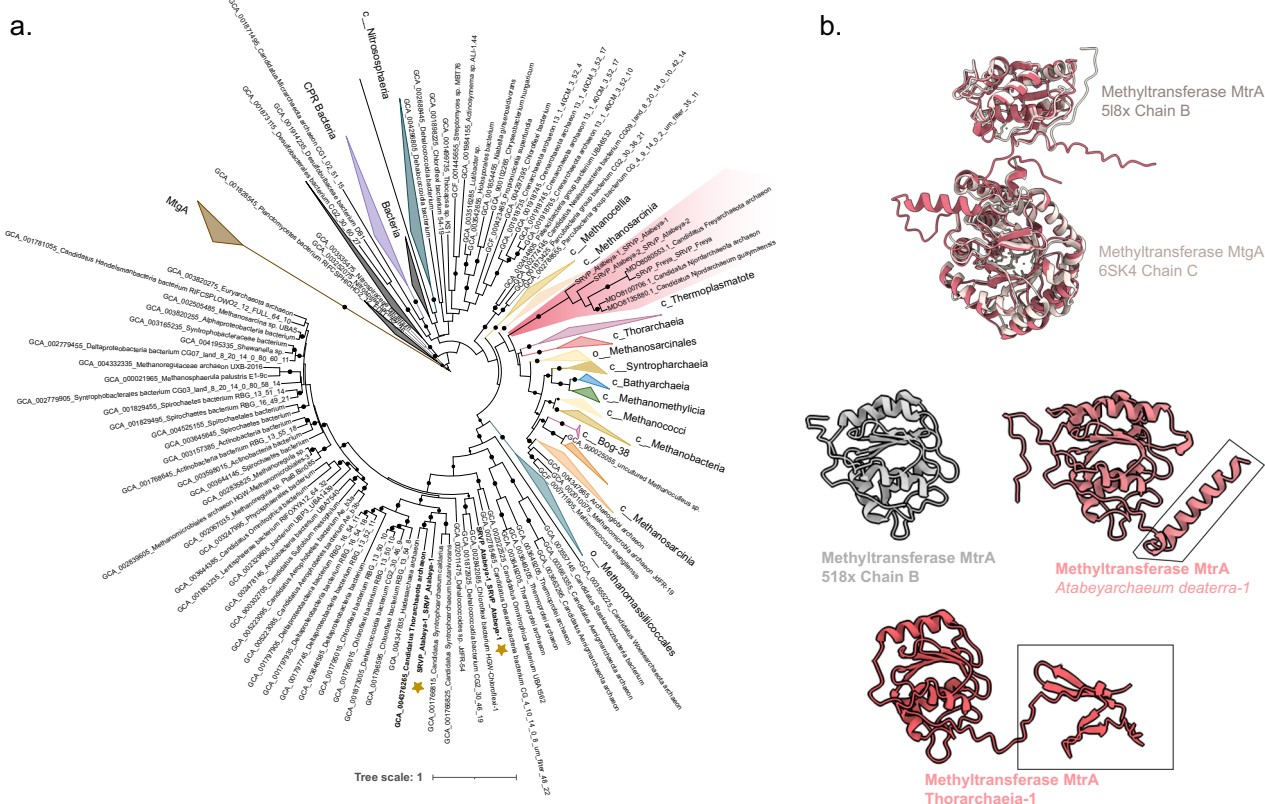

**Fig. 5 | Non-methanogenic MtrA, MtrH and MtrAH fusion methyltransferases.**
**a** Maximum likelihood phylogeny of MtrA and the MtrAH fusion, with reference to Tetrahydromethanopterin S-methyltransferase subunit A (MtrA) with the closest corresponding domains being MtrA from the characterized Tetra-hydromethanopterin S-methyltransferase subunit A (MtrA) protein (PDB ID: 5L8X)[105]. The coral-colored clade is the novel fusion present in Atabeyarchaeia, Freyarcheia, and other Asgardarchaeota members. **b** AlphaFold models of Atabeyarchaeia-1 MtrAH (fusion) in coral aligned with the gray corresponding

domains of the characterized protein Tetrahydromethanopterin S-methyltransferase subunit A (MtrA) (PDB ID: 5L8X)[106] and Methyltransferase (MtgA) from *Desulfitobacterium hafniense* in complex with methyl-tetrahydrofolate (PDB ID: 6SK4) at the N terminus. We also modeled the putative MtrA present in Atabeyarchaeia-1 with the closest corresponding domains being MtrA from the characterized Tetrahydromethanopterin S-methyltransferase subunit A (MtrA) protein (PDB ID: 5L8X).

Data 9). Both Atabeyarchaeia and Freyarchaeia genomes encode tri-methylamine methyltransferase MttB (COG5598). Phylogenetic analysis suggests that MttB (Supplementary Fig. 12) and MtbC (Supplementary Fig. 13) belong to a previously uncharacterized group of methyltransferases, similar to those found in Njordarchaeales, Helarchaeales, Odinarchaeia and TACK members, including Brock-archaeia and Thermoproteota. In methanogens that encode *mttB*, this gene has an amber codon encoding the amino acid pyrrolysine in the active site[41,42]. The archaea from this study do not encode pyrrolysine, suggesting Freyarchaeia and Atabeyarchaeia encode a non-pyrrolysine MttB homolog, likely a quaternary amine (QA) dependent methyltransferase[43]. Only a fraction of QA methyltransferase sub-strates have been identified, and these include glycine betaine, proline betaine, carnitine, and butyrobetaine[43–46]. The methyl group from the QA may be transferred to THF or $H_4$MPT branches of the WLP, akin to the mechanisms described in archaea with the capacity for non-methanogenic anaerobic methylotrophy, including Freyarchaeia, Sifarchaeia, Brockarchaeia, and Culexarchaeia[11,12,47,48]. Consumption of QA compounds may reduce the pool of potential substrates for methanogenic methane production.

We also identified genes in the Freyarchaeia genome encoding a putative molybdenum-containing hydroxylase distantly related to the aerobic carbon monoxide dehydrogenase complex (CoxLMS). Phylogenetic analysis places the large subunit protein (related to CoxL) in a monophyletic group with other archaea including

Thermoplasmatales, Marsarchaeota, and Culexarchaeia (Supple-mentary Fig. 14a). Analysis of the protein sequence reveals that the putative large-subunit aerobic CO dehydrogenases (CoxL) are miss-ing the characteristic VAYRCSFR motif, which is critical for CO binding in the form I Cox proteins[49,50]. The genetic organization suggests the Freyarchaeia enzyme is structurally similar to carbon monoxide dehydrogenase, but likely oxidizes distinct substrate (Supplementary Fig. 14b). Consistently, other members of the molybdenum-containing hydroxylase superfamily oxidize other substrates, such as aldehydes or purines[48,50,51].

## Carbon compound metabolic pathways

There are indications that Freyarchaeia and Atabeyarchaeia display distinct metabolic preferences for various soil carbon compounds (Fig. 2 and Supplementary Fig. 15). Freyarchaeia exhibits a genetic repertoire to break down various extracellular lignin-derived com-pounds, including 5-carboxyvanillate. Other substrates that we predict can be metabolized by Freyarchaeia carbohydrate-active enzymes include hemicellulose (C5), cellobiose, maltose, and cellulose (C12). We predict that cellodextrin (C18) compounds can be converted to glucose via beta-glucosidase (BglBX). The findings implicate Freyarchaeia in the metabolism of plant-derived soil carbon com-pounds. Glucose, resulting from the degradation of complex carbo-hydrates, as well as ribulose and other carbon substrates, likely enters the modified Embden-Meyerhof-Parnas (EMP) pathway, yet the genes

of this EMP pathway genes are only weakly expressed (Supplementary Fig. 15, Supplementary Data 6 and Supplementary Data 8). In addition, Freyarchaeia encode and express an array of genes for the uptake of carbohydrates, including sugar transporters from both the major facilitator and ABC superfamilies, suggesting an active role in efficiently assimilating diverse carbon substrates from soil environments Atabeyarchaeia also harbor genes of the EMP glycolytic pathway, producing ATP through the conversion of acetyl-CoA to acetate (Supplementary Fig. 15). Unlike Freyarchaeia, which likely feed glucose into the EMP pathway, the entry point for Atabeyarchaeia to the EMP pathway appears to be fructose 6-phosphate (F6P). This is relatively uncommon for Asgard archaea but is reminiscent of the pathway in *Helarchaeales*[7], an order of Lokiarchaeia. We identified Atabeyarchaeia transcripts for all but one of the genes for the steps from G6P to acetate (Supplementary Data 6).

Atabeyarchaeia and Freyarchaeia use different enzymes to produce pyruvate. Atabeyarchaeia encodes the oxygen-sensitive reversible enzyme, pyruvate:phosphate dikinase (PpdK), whereas Freyarchaeia encodes unidirectional pyruvate water dikinase/phosphoenolpyruvate synthase (PpS) and pyruvate kinase (Pk), producing phosphoenolpyruvate and pyruvate[52], respectively. Pyruvate generated via the EMP pathway can then be converted to acetyl-CoA by pyruvate:ferredoxin oxidoreductase (PorABCDG) complex using a low-potential electron acceptor such as a ferredoxin. Alternatively, acetyl-CoA can also be generated via pyruvate formate-lyase (*pflD*) generating formate as a byproduct. The final step involves the conversion of acetyl-CoA to acetate via acetate-CoA ligase (ADP-forming), producing ATP via substrate-level phosphorylation, in a crucial energy-conserving step during the fermentation of carbon compounds in both lineages.

Lacking the ability to phosphorylate C6 carbon sources, Atabeyarchaeia is predicted to convert ribulose-5-phosphate (C5) and fix formaldehyde (C1) into hexulose-6-phosphate (H6P) via the ribulose monophosphate (RuMP) and non-oxidative pentose phosphate (NO-PPP) pathways (Fig. 2 and Supplementary Fig. 16). The Atabeyarchaeia RuMP pathway bifunctional enzymes (HPS-PH and Fae-HPS) are common in archaea and similar to methylotrophic bacterial homologs[53]. The RuMP pathway in these Asgard archaea can modulate the formaldehyde availability, a byproduct of methanol oxidation, microbial organic matter decomposition, and combustion. High expression of aldehyde-ferredoxin oxidoreductases (AOR) genes suggests another mechanism for the interconversion of organic acids to aldehydes. For example, aldehyde detoxification (e.g., formaldehyde to formate) and source of acetate from acetaldehyde. Atabeyarchaeia-1, Atabeyarchaeia-2, and Freyarchaeia encode multiple AOR gene copies 5, 6, and 8 respectively. Phylogenetic analyses (Supplementary Fig. 17) suggest that both Asgard lineages encode AOR genes related to the FOR family that oxidize C1-C3 aldehydes or aliphatic and aromatic aldehydes (e.g., formaldehyde or glyceraldehyde)[54–56]. Furthermore, Freyarchaeia also encodes a tungsten-based AOR-type enzyme (XOR family) found in cellulolytic anaerobes with undefined substrate specificity[57] (Supplementary Fig. 17). Of the classified AORs, only one gene is expressed in Atabeyarchaeia-2 (Fig. 3). Yet, some of the unclassified AOR genes are among the most expressed genes (>10 transcripts) in the Atabeyarchaeia genomes. Despite the lack of biochemical characterization for most AOR families, these observations suggest a key role of multiple aldehydes in the generation of reducing power in the form of reduced ferredoxin.

Similar to other Asgard archaea[7,10,27], Atabeyarchaeia and Freyarchaeia encode genes for the large subunit of type IV and methanogenic type III Ribulose 1,5-bisphosphate carboxylase (RbcL) (Supplementary Fig. 18) a key enzyme in the partial nucleotide salvage pathway. This pathway facilitates the conversion of adenosine monophosphate (AMP) to 3-phosphoglycerate (3-PG), potentially leading to further metabolism into acetyl-CoA[58].

Anaerobic glycerol (C3) metabolism by Atabeyarchaeia and Freyarchaeia is predicted based on the presence of glycerol kinase (GlpK), which forms glycerol-3-phosphate (3PG) from glycerol. 3PG (along with F6P) can be broken down via the EMP pathway, or 3PG can be converted to dihydroxyacetone phosphate (DHAP) via GlpABC. DHAP can also serve as a precursor for sn-glycerol-1-phosphate (G1P), the backbone of archaeal phospholipids. Freyarchaeia have an extra GlpABC operon, the GlpA subunit of which clusters phylogenetically with GlpA of Halobacteriales, the only known archaeal group capable of glycerol assimilation (Supplementary Fig. 19).

All three genomes have a partial TCA cycle similar to other anaerobic archaeal groups such as methanogens[59]. They encode succinate dehydrogenase, succinyl-CoA synthetase, and 2-oxoglutarate ferredoxin reductase, which are important intermediates for amino acid degradation (e.g., glutamate). Only Atabeyarchaeia can convert fumarate to malate via fumarate hydratase. The only portion of the TCA cycle transcribed in any genome is 2-oxoglutarate/2-oxoacid ferredoxin oxidoreductase, which can produce reducing power in the form of NADH.

A clue suggesting that amino acids are an important resource for Atabeyarchaeia and Freyarchaeia is the high expression of genes for protein and peptide breakdown (Fig. 2). All three organisms are predicted to have the capacity to break down fatty acids via beta-oxidation including crotonate (short-chain fatty acid) via the poorly described crotonate pathway. Furthermore, they encode some enzymes involved in fermenting amino acids to H +, ammonium, acetate, and NAD(P)H via the hydroxyglutarate pathway (Supplementary Data 7). The genomes also encode amino acid transporters and these are also transcribed in both archaeal groups. The ability to anaerobically degrade amino acids is consistent with predictions of the metabolism of the last Asgard common ancestor[4,9].

In addition, Freyarchaeia and Atabeyarchaeia can reverse the step in the formyl branch of the WLP that transforms glycine into methylenetetrahydrofolate (methylene-THF). Methylene-THF may then be converted to methyl-THF and then to formyl-THF, producing reducing power (Fig. 2). Ultimately, the methyl group may be used to form acetate via the WLP. Interestingly, Atabeyarchaeia-2 and Freyarchaeia expressed methylenetetrahydrofolate reductase (MTHFR), which is homologous to the enzyme used in the bacterial WLP and also plays a role in folate biosynthesis.

## Environmental protection and adaptations

We predict that both Atabeyarchaeia and Freyarchaeia are anaerobes that express genes encoding oxygen-sensitive enzymes and proteins that have evolved to protect against oxidative and other environmental stressors. Notably, all three organisms in our study encode an ancestral version of clade I catalases (KatE), (Supplementary Fig. 20). Additionally, their genomes encode Fe-Mn superoxide dismutase (SOD2), which plays a critical role in the dismutation of superoxide radicals into less harmful molecular oxygen and hydrogen peroxide.

Freyarchaeia encodes a catalase-peroxidase (WXG42230.1) that shares 88% protein identity with a catalase from Methanosarcina (WP_048128959.1). This protein clusters within the Methanomicrobia (Supplementary Fig. 21), suggesting acquisition via lateral gene transfer. Previous analyses have described these enzymes in acetogenic and sulfate-reducing bacteria and methanogenic archaea, but to our knowledge, not in Asgard archaea. As this enzyme does not occur in Freyarchaeia from other environments, it may be an adaptation to soil[60].

We infer that Atabeyarchaeia and Freyarchaeia use selenocysteine (Sec), the 21st amino acid, due to the presence of the Sec-specific elongation factor and Sec tRNA in their genomes. Additional Sec components, including phosphoseryl-tRNA kinase (Pstk), Sec synthase (SecS), selenophosphate synthetase (SPS) genes, and multiple Eukaryotic-like Sec Insertion Sequences are also present

(Supplementary Data 10). Phylogenetic analysis shows that the Sec elongation factor sequences from Atabeyarchaeia and Freyarchaeia are closely related to other Asgard members and Eukaryotes (Supplementary Fig. 22). We identified multiple selenoproteins encoded within each genome, including CoB-CoM heterodisulfide reductase iron-sulfur subunit (HdrA), peroxiredoxin family protein (Prx-like), selenophosphate synthetase (SPS), and the small subunit (~ 50 aa) of NiFeSec (VhuU). In VhuU, Sec plays a crucial role in mitigating oxidative stress[61]. Sec can also enhance the catalytic efficiency of redox proteins[62], and the identified selenoproteins have the characteristic CXXU or UXXC sequence (Supplementary Data 11) observed in redox-active motifs[63].

## Discussion

Here, we reconstructed and validated three complete Asgard archaeal genomes from wetland soils in which these archaea comprise less than 1% of complex microbial communities. We used these genomes to define their chromosome lengths, structure, and replication modes. It is relatively common for authors to report circularized genomes as complete, but this may be erroneous due to the prominence of local assembly errors, chimeras, scaffolding gaps, and other issues in de novo metagenome assemblies[22,64]. Our genomes were thoroughly inspected, corrected, and vetted after circularization, steps previously described to complete genomes from metagenomes[65]. These complete genomes are one of the first manual curations of short-read metagenomic data verified entirely with long-read analysis (Oxford Nanopore and/or PacBio) and the first complete short-read environmental Asgard genomes. Two of these genomes are from Atabeyarchaeia, a previously undescribed Asgard group and the first complete genome for Freyarchaeia. The predicted bidirectional replication in Freyarchaeia and Atabeyarchaeia suggests the likelihood of this DNA replication method in the last Asgard-Eukaryotic common ancestor. Additional high-quality Asgard genomes are needed to determine the precise evolutionary role of replication in the emergence of the complex cellular organization characteristic of eukaryotes.

Overall, prior studies primarily predict that Asgard archaea degrade proteins, carbohydrates, fatty acids, amino acids, and hydrocarbons[5,6,10,66]. *Lokiarchaeales*, Thorarchaeia, Odinarchaeia, and Heimdallarchaeia are primarily organoheterotrophs with varying capacities to consume and produce hydrogen[27]. *Helarchaeales* are proposed to anaerobically oxidize hydrocarbons[7,10,67], whereas Freyarchaeia and Sifarchaeia are predicted to be heterorganotrophic acetogens capable of utilizing methylated amines[11,12]. Hermodarchaeia is proposed to degrade alkanes and aromatic compounds via the alkyl/benzyl-succinate synthase and benzoyl-CoA pathway[10]. *Gerdarchaeales* may be facultative anaerobes and utilize both organic and inorganic carbon[8]. Wukongarchaeia is predicted to be chemolithotrophic acetogens, encoding the WLP and ADP-dependent acetyl-CoA synthetase in the absence of the TCA cycle. Atabeyarchaeia and Freyarchaeia share several metabolic pathways with new lineages from the Asgard sister-clade TACK (e.g., Brock- and Culexarchaeia) and other deeply branching Asgard lineages. Based on the genomic and metatranscriptomic analyses, we predict that the soil-associated Atabeyarchaeia and Freyarchaeia are chemoheterotrophs that likely degrade amino acids and other carbon compounds. Both encode the EMP Pathway for cellular respiration and the WLP for $CO_2$ fixation.

Although Atabeyarchaeia and Freyarchaeia share key central metabolic pathways, they differ in that Freyarchaeia can metabolize compounds such as formaldehyde (C1), glycerol (C3), ribulose (C5), and glucose (C6), whereas Atabeyarchaeia can only metabolize C1, C3 and C5 compounds (Fig. 2). The ability to metabolize C3 and C5 compounds is rare in Asgard archaea. Both lineages possess the genetic repertoire necessary for converting carbohydrates into acetate and may be capable of growth as anaerobic acetogens via the WLP.

Our results suggest acetogenesis is a common metabolic feature shared by marine sediment and wetland soil-associated Asgardarchaeota. The soil metagenomes described here showed potential for acetogenesis and homo-acetogenesis through WLP. Supporting previous analysis indicates acetogenesis is an ancestral trait within this phylum[4]. Homoacetogenic pathways have also been described in several marine Asgard lineages, including Lokiarchaeia, Sifarchaeia, Thorarchaeia, and Wukongarchaeia[3,5,6,11,27,36]. We were able to identify both acetogenesis and homo-acetogenesis genes in the two lake sediment Atabeyarchaeia MAGs and Freyarchaeia MAGs extracted from deep-sea (Guaymas Basin) and hot spring sediments (Tibet Plateau, Yunnan Province, and Radiata Pool)[4,12]. The ability to fix $CO_2$ and produce acetate via the WLP could provide a competitive advantage for Asgard archaea in marine and terrestrial environments, where competition may limit labile organic carbon sources. Acetate production in today's Asgard archaea from various environments may support the growth of other microbial community members, including acetoclastic methanogens.

Similar to other Asgard archaea, they have methyltransferase complexes potentially involved in the catabolism of quaternary amines (or yet unknown methylated substrates). These methylotrophic Asgard archaea may compete with methanogens and other anaerobic methylotrophic groups (Brockarchaeia and Culexarchaeia) identified within these California soil samples for methylated compounds (Fig. 6). These results align with recent studies suggesting a broader presence of methylotrophic metabolisms among archaea[10,47,48]. It also opens up avenues for exploring the environmental impact of these metabolisms, particularly in relation to carbon cycling and greenhouse gas emissions[68].

Of particular interest is the predicted metabolic capability of Atabeyarchaeia and Freyarchaeia to degrade aldehydes. Aldehydes in soils come from several sources, including the microbial breakdown of methanol potentially produced from methane oxidation, degradation of plant and animal compounds, and products of industrial combustion and wildfires (e.g., volatile organic compounds). In fact, the California wetland soil that hosts these archaea contains charcoal, likely produced by wildfires. They are also predicted to be capable of growing on glycerol under anaerobic conditions capacity previously undescribed in Asgard archaea.

The growth of soil-associated Atabeyarchaeia and Freyarchaeia could be supported by hydrogen oxidation in tandem with the Wood-Ljungdahl pathway (WLP), as evidenced by the expression of genes for group 3c [NiFe]-hydrogenase and the WLP. Given that both sets of genes were actively expressed in situ, we infer that the growth of these Asgard archaea reduces the concentration of $H_2$ available for methane production by coexisting methanogens. The presence of syntenic blocks encoding heterodisulfide reductase complexes, [NiFe] hydrogenases and ATP synthase suggests a sophisticated apparatus for energy transduction, resembling mechanisms previously characterized in other archaeal groups[35]. In addition, our results suggest the existence of an electron bifurcation mechanism in both Asgard archaea lineages, where electrons can be transferred from $H_2$ to ferredoxin and an unidentified heterodisulfide intermediate[27]. Atabeyarchaeia and Freyarchaeia also have membrane-bound group 4 [NiFe]-hydrogenases that likely facilitate the oxidation of reduced ferredoxin generated through fermentative metabolism. However, this complex is novel in that it includes an HL helix on the L-like subunit and two antiporters, neither of which are present in other structurally characterized group 4 respiratory hydrogenases. The functional modeling of these complexes reveals structural congruences with known respiratory enzymes, hinting at a potential for chemiosmotic energy conservation that may be a widespread feature among the Asgard clade. The findings indicate a potential evolutionary connection between hydrogenases and complex I, aligning with the hypothesis that complex I may have evolved from ancestral hydrogenases[31,69].

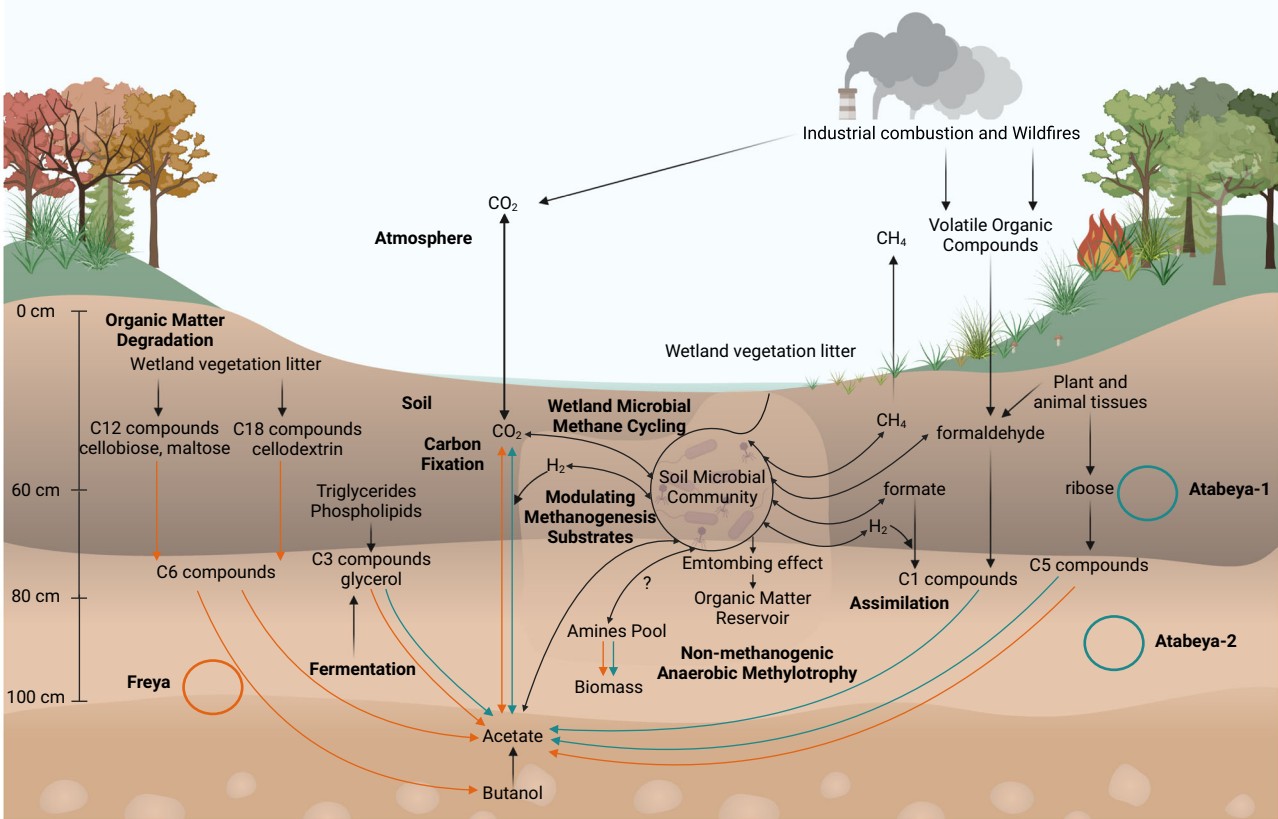

**Fig. 6 | Overview of the wetland soil dynamics and biogeochemical cycling in Atabeyarchaeia and Freyarchaeia.** Complete genomes for Atabeyarchaeia and Freyarchaeia are shown with green and orange circles, respectively. The Atabeyarchaeia (Atabeya-1 and Atabeya-2) and Freyarchaeia (Freya) genomes were isolated and carefully curated and closed from wetland soil between 60-100 cm. These anaerobic lineages were shown in this study to encode the Wood-Ljungdahl Pathway for $CO_2$ fixation (e.g., methylated compounds such as quaternary amines) and Embden-Meyerhof-Parnas (EMP) Pathway, components of chemolithotrophy and heterotrophy, producing acetate shown by the arrows (green and orange), corresponding to the taxonomic lineage colors. The metabolic versatility of these soil Asgardarchaeota like their marine counterparts may provide a competitive advantage in wetlands. A detailed description of the specific pathways is found in the main text, Fig. 2, and supplementary materials. Created using BioRender.com.

We manually curated three complete genomes for Asgard archaea from wetland soils, uncovering bidirectional replication and an unexpected abundance of introns in tRNA genes. These features suggest another facet of the evolutionary relationship between archaea and eukaryotes. Metabolic reconstruction and metatranscriptomic measurements of in situ activity revealed a non-methanogenic, acetogenic lifestyle and a diverse array of proteins likely involved in energy conservation. The genome analysis uncovered some metabolic similarities between soil and sediment-associated Asgard archaea[9,27]. In addition, the unusual genomic features, such as high intron prevalence and the presence of ESPs call for deeper investigations into their biological significance and contribution to the adaptability and evolution of Asgard archaea. Overall, the findings point to metabolic flexibility and adaptation to soil conditions of wetlands. Finally, they contribute to the cycling of carbon compounds that are relevant for methane production by coexisting methanogenic archaea. Future studies should aim to explore the functional roles of the identified genes and pathways in situ, which will further elucidate the ecological impacts and evolutionary history of these enigmatic microorganisms.

## Methods

### Sample acquisition, nucleic acid extraction, and sequencing

We collected soil cores from a seasonal vernal pool in Lake County, California, in October 2018, October 2019, November 2020, and October 2021. Samples were frozen in the field using dry ice and kept at − 80 C until extraction. The Qiagen PowerSoil Max DNA extraction kit was used to extract DNA from 5-10 g of soil, and the Qiagen AllPrep DNA/RNA extraction kit was used to extract RNA from 2 g of soil. Samples were sequenced by the QB3 sequencing facility at the University of California, Berkeley on a NovaSeq 6000. Read lengths for the 2018 DNA samples, and the RNA samples were $2 \times 150$ bp and $2 \times 250$ bp for the 2019–2021 DNA samples. A sequencing depth of 10 Gb was targeted for each of the 2018, 2020, and 2021 samples, and 20 Gbp for each of the 2019 samples. A subset of deep soil samples from 2021 were sequenced using PacBio and Oxford Nanopore technologies. For Oxford Nanopore sequencing, the Ligation Sequencing Kit (LSK114) was used to prepare native DNA libraries and the Qiagen Repli-G mini kit and LSK114 were used to prepare amplified DNA libraries. Samples were sequenced on FLO-PRO114M flowcells on the PromethION for 72 hours.

### Sample preparation and metatranscriptomic analysis

To address potential technical variations and ensure robustness in our findings, we implemented several strategies during sample preparation. Homogenization of soil samples: Prior to RNA extraction, soil samples were thoroughly homogenized to ensure the representativeness of each environmental sample, which is crucial for minimizing variability due to sample heterogeneity. Pooling of Samples: Given the challenges in obtaining sufficient RNA from deep soil samples, we pooled replicate samples before sequencing. This not only facilitated adequate RNA yields for robust metatranscriptomic analysis but also helped average out individual sample variability, thereby reducing the

impact of technical variation on our results. Despite the absence of technical replicates, these preparatory measures should help minimize technical variation.

### Illumina metagenomic assembly, binning, and annotation and genome statistics

Metagenomic sequencing reads were trimmed for adapter sequences and quality using sickle[70]. The filtered metagenomic reads from 2018 were assembled using the IDBA_UD assembler[71]. The reads from 2019-2020 using metaSPAdes[72] Contigs greater than 2.5 Kb were retained and sequencing reads from all samples were cross-mapped against each resulting assembly using Bowtie2[73]. The resulting differential coverage profiles were filtered at a 95% read identity cutoff and then used for genome binning with MetaBAT2[74], VAMB[75], and MaxBin2[76]. The resulting genome bins were assessed for completeness and contamination using CheckM2[77] and were manually curated using taxonomic profiling with GGKBase (www.ggkbase.berkeley.edu). The first iteration of taxonomy was assigned to genome bins with GTDB-Tk v2.3.0[78] and further validated with phylogenetic trees of single-copy marker genes. We also downloaded all the publicly available Asgard genomes from BV-BRC and used CheckM2 to estimate genome completeness.

### PacBio metagenomic sequencing and assembly

Samples from September 9, 2021, from 140 cm and 75 cm, were sequenced using Sequel II to generate PacBio HiFi reads. Reads were quality trimmed using BBDuk (bbduk.sh minavgquality = 20 qtrim = rl trimq = 20)[79] and assembled with hifiasm-meta[80].

### Oxford Nanopore metagenomic assembly

Reads were base called with Guppy using the dna_r10.4.1_e8.2_sup@v3.5.1 model. Reads were filtered for an average quality >10 and a minimum length of 1 kb using fastp (v0.23.2). Adapters were trimmed using porechop (v0.2.4). Branching artifacts caused by multiple displacement amplification were removed by aligning amplified reads to themselves with mappy (v2.24). Reads with non-diagonal self-alignment were removed. Both amplified and native reads were jointly assembled with metaFlye (v2.9)[81], long-read polished with medaka consensus (v1.7.1), and short-read polished with Hapo-G (v1.3.1).

### Manual genome curation of short-reads genomes and validation using long-reads

The manually curated genomes were de novo reconstructed from high-quality Illumina metagenomic data as described previously[22]. From soil samples taken at various depths, we recovered draft Illumina-based MAGs corresponding to Atabeyarchaeia-1, Atabeyarchaeia-2, and Freyarchaeia. The curation process involved the identification and removal of obvious chimeric regions, which were indicated by abrupt changes in GC content or by insufficient Illumina read mapping support. We also corrected sequences in regions with imperfect read alignment, allowing no single nucleotide polymorphisms (SNPs), by mapping reads at a reduced stringency threshold (allowing for up to 3% SNPs). This was followed by manual curation of the consensus sequence, including insertion, deletion, or substitution of individual base pairs. The extension of contig ends was conducted using unplaced Illumina reads. High read coverage was interpreted as indicative of the genomic termini. A genome was deemed complete when it displayed uninterrupted support from Illumina reads. The final assessment of genome completeness was performed by examining the cumulative GC skew and ensuring alignment with known complete genomes from related taxa. The Average Amino Acids of the new genomes were performed using AAI: Average Amino acid Identity calculator tool (http://enve-omics.ce.gatech.edu/aai/) and using compareM (v.0.0.23) with the 'aai_wf' at default settings (https://github.com/dparks1134/CompareM). Replichores of complete genomes were

predicted according to the GC skew and cumulative GC skew calculated by the iRep package (gc_skew.py).

### Metabolic pathway reconstruction of complete genomes and phylogenetic analysis of key functional genes

We utilized Prodigal (v2.6.3) to predict the genes for Atabeyarchaeia and Freyarchaeia genomes in this study[82]. A suite of databases, including KofamKOALA (v1.3.0)[83] and InterProScan (5.50-84.0)[84] were combined to begin annotation. Complete genomes were also annotated with HydDB[85], METABOLIC (v4.0)[86], PROKKA (v1.14.6), DRAM[87], and MEBS (v2.0). Our methods also incorporated a microbial genome annotation workflow as previously described in the Undinarchaeota study[88].

For each gene/protein of interest, references were compiled by 1) BLASTing the corresponding gene against the NCBI nr database and their top 50 hits clustered by CD-HIT using a 90% similarity threshold, 2) obtaining sequences from the high-quality manually annotated UniProtKB/Swiss-Prot/InterPro/PFAM (reviewed) (*catalases*), or 3) using previously published alignment and creating hmm models to extract Atabeyarchaeia and Freyarchaeia sequences (*AOR, catalases, and MttB-like homologs*). The final set genes/proteins were aligned using MAFFT v.7.407 (*AOR and catalases*)/v7.490 (*ADH and MttB-like homologs*), trimmed with trimAl v1.4.rev15 (*AOR, ADH, catalases,* and *MttB-like homologs*), and a phylogenetic tree was inferred using IQ-TREE v.1.6.6 (*AOR and catalases*)/v2.0.7 (*ADH and MttB-like homologs*) using automatic model selection. More detailed descriptions for each phylogeny, including model selection are discussed in the supplemental figure captions.

### Environmental distribution of Atabeyarchaeia and Freyarchaeia

The analysis of metagenomic data to determine the presence and relative abundance of LC30 and Jordarchaeia was performed using the Sandpiper tool, which is available at https://sandpiper.qut.edu.au/. Sandpiper was utilized to search across a wide range of publicly available metagenomes to identify the occurrences of these organisms. The tool provides a streamlined workflow for uploading metagenomic data, selecting specific targets (in this case, Atabeyarchaeia and Freyarchaeia), and analyzing the data based on relative abundance and environmental metadata.

### Identification of [NiFe]-hydrogenase and phylogenetic analyses

We extracted [NiFe]-hydrogenase sequences from complete genomes as well as publicly available Asgardarchaeota genomes from NCBI and BCWT using custom Hidden Markov Models (HMMs). To confirm the accuracy of the candidate [NiFe]-hydrogenase sequences, we checked the identified sequences by looking at nearby genes and ensuring the presence of essential hydrogenase accessory genes and confirmed the preliminary classification using HydDB[85]. We combined our [NiFe]-hydrogenase sequences with those from the HydDB database. Then, we aligned all the sequences using MAFFT (parameters:−localpair −maxiterate 1000). After alignment, we looked for the N-terminal and C-terminal CxxC conserved motifs. We then used TrimAl v (parameters: −gt 0.5) to clean up the alignment, removing parts where more than 50% of the sequences had gaps. We used IQ-TREE version 1.6.12 with the best-fit model according to the Bayesian information criterion (BIC) and ultrafast 1000 bootstrap method to estimate support values for the tree branches. The output tree was visualized using ItoL, and the subgroups were classified based on known HYDB references[85].

### Identification of Selenocysteine machinery and Selenoproteins, tRNA, and intron predictions

The Sec machinery (pstK, EFsec, SecS, and SPS) was identified using HMMER v3.3.1[89] via hmmscan against the TIGRFAM v15.0 database[90] and also confirmed via Selenoprofiles v4.4.8[91], using the -p machinery option. Selenoprofiles v4.4.8 and the Seblastian-SECIS3 software[92] was

accessed to identify potential selenoproteins, which were manually curated via alignment to known selenoproteins as previously reported[93]. SECIsearch3 was used to identify SECIS for selenoproteins.

For identification of predicted pre-tRNA, we used tRNAscan-SE v.2.0.12 (default settings and -A for archaea model)[94]. The R2DT software[95] was used to predict and visualize the tRNA secondary structure, which also led to the identification of additional introns that were not detected by tRNAscan. We identified Sec tRNA with the Secmarker v0.4 webserver[96] with a minimum Infernal score of 35.

## Metatranscriptome sequencing and analysis

A subset of the deep SRVP soil samples (60 cm, 70 cm, 80 cm, and 100 cm) were used for metatranscriptomics. Asgard genomes were first dereplicated using deep v 3.4[97], resulting in a set of representative genomes. A Bowtie2 index was constructed from the dereplicated genomes to expedite the sequence alignment process. Post alignment, a custom-built Python script[98] was utilized to scrutinize the paired reads from the alignment (BAM) files and deliver statistical insights on gene expression. This script initiates by creating a Pysam 'Alignment-File' object, enabling a systematic traversal through the '.bam' files by parsing the complex binary alignments. It then proceeds to iterate over each gene listed in a predefined set. For each gene, the script tallies the reads that satisfy user-defined thresholds for Average Nucleotide Identity (ANI) and Mapping Quality (MAPQ). These thresholds, designed to uphold a high level of alignment quality and sequence identity, are critical in enhancing the reliability of the data. Upon the completion of this process, the script outputs the filename, the gene name, and the count of reads that conform to the defined parameters. This pipeline permits an in-depth investigation of RNA hits within the dereplicated genomes, thereby illuminating gene expression profiles across diverse environmental scenarios.

To quantify the number of reads per gene, we further utilized a Python script (filter_counts.py). This script operates based on gene predictions produced by the Prodigal software for the genomes of interest. With the required gene predictions secured through a rerun of Prodigal, the script was set in motion on the BAM files. The command executed was as follows: python filter_counts.py -q 10 -m 0.97 derep_genomes.faa. This command sets the minimum MAPQ score (-q 10) and the minimum ANI (-m 0.97) to ensure high-quality alignments and a high degree of sequence identity. This adherence to stringent criteria bolsters the reliability and robustness of our analytical process.

Mapped mRNA reads were standardized by the calculation of reads per kilobase of transcript per million reads mapped (RPKM), which allows for the direct comparison of the level of transcription of all genes in the de novo assembly (Supplementary Data 6). RPKM is calculated in Rockhopper as RPKM = ((number of mapped reads/length of transcript (gene) in kilobase)/million mapped reads))[99].

## Abundance and distribution of archaea

The dereplicated Atabeyarchaeia, Freyarchaeia, and other archaeal genomes present in the 36 metagenome samples were mapped to all the metagenome reads independently using BBMap[79] (parameters: nodisk = t pigz = t unpigz = t ambiguous = random). We set the minimum identity for mapping to 0.9 and handled ambiguous mappings randomly. The percentage of relative abundance of each genome and the unmapped read percentage values were calculated using coverM (parameters: -m relative_abundance –min-read-percent-identity 95).

## Phylogenetic analyses of Asgard genomes

We used three different sets of markers for the phylogenetic analysis. 47 arCOGs (Archaeal cluster of orthologous genes), a subset of previously described marker set[100], were extracted with hmmsearch (v3.1b2) from the 9 MAGs in this study, 303 additional Asgard MAGs, and 36 TACK publicly available MAGs. Korarchaeales MAGs were

excluded from the analysis due to the hyperthermal-sensitive marker genes[4]. We chose to exclude arCOG01183 from the previous set as it was found in less than 50% of the MAGs. mafft (v7.310) and BMGE (v1.12) were used to align and trim the concatenated alignments. Maximum likelihood phylogenetic trees were generated using IQ-TREE (v1.6.1) to test various models and topologies to obtain bootstrap values and the LG + F + R10 model was selected (Fig. 1c). In addition, we compared this phylogeny with a set of 37 single-copy marker genes from Phylosift (v1.0.1)[101]. The sequences were manually accessed, aligned with mafft auto (v7.490), trimmed with BMGE (v2.0), and a maximum likelihood phylogeny generated with IQ-TREE (v2.0.7), using the LG + F + R10 model (Supplementary Fig. 25). All marker sets resulted in the same phylogeny for the 9 added MAGs. 16S ribosomal proteins were extracted from 439 Asgard and 39 TACK MAGs with barrnap 0.9, using options –kingdom arc –lencutoff 0.2 –reject 0.3 –evalue 1e-05. Sequences were manually curated, aligned with mafft auto (v7.490) and masked with Geneious Prime, and ran with IQ-Tree (v2.0.7). 16 S phylogeny separates Atabeyarchaeia MAGs from other Asgard clades (Supplementary Fig. 2). Amino acid identity (AAI) was determined with CompareM (v0.0.23) to distinguish taxonomic level. We assigned numbers from 1-9 as candidate groups within this new Asgardarchaeota class.

## Identification and analysis of genomic ESPs

PSI-BLAST was used to query the Asgard proteomes against the AsCOGs (Asgard database) and arCOGs databases[3]. Hits were only considered if the absolute difference between the subject and query sequence was less than 75% of the length of the query sequence. Hits were then dereplicated by taking the best hit for each query.

## Protein structure prediction using AlphaFold-multimer and ColabFold

Structural predictions for group 4 [NiFe]-hydrogenases, CoxLMS, MtrA, and MtrAH fusion proteins were performed using ColabFold, incorporating AlphaFold-multimer capabilities[102]. These predicted structures were then visualized and aligned using ChimeraX[103], with corroborating reference structures obtained from the Protein Data Bank[104]. This structural modeling was integral in reinforcing our phylogenetic and metabolic inference.

## Nomenclature, etymology, and proposal of type material

**Description of Atabey- taxonomy.** Based on the phylogenetic placement and metabolic traits uncovered in our research, we propose new taxonomic nomenclature for the identified Asgardarchaeota organisms. The chosen nomenclature encapsulates various facets that resonate with the organisms' nature and their ecological habitat. The term 'Atabey' is a homage to the Mother Goddess, often referred to as the Earth Mother in the Taíno mythology, symbolizing fertility, abundance, and the nurturing aspects of nature. We suggest the taxonomic hierarchy of class 'Ca. Atabeyarchaeia', order 'Ca. Atabeyarchaeales', family 'Ca. Atabeyarchaeaceae', genus 'Ca. Atabeyarchaeum', and species 'Ca. *Atabeyarchaeum deaterra*', which have been registered via SeqCode (seqco.de/r:igzul8qd).

**Description of Atabeyarchaeum deiterre sp. nov.** (de.i.ter'rae L. fem. n. *dea*, deity; L. fem. n. *terra*, Earth; N.L. gen. n. *deiterrae*, of an Earth deity)

A soil-associated uncultured lineage, represented by the complete genome "Atabeyarchaeia-1" (2.81 Mbp), NCBI BioSample SAMN40176692 in BioProject PRJNA1050611, recovered from California soil. According to SeqCode Registry v1.0.3, the submitted complete genome passes data quality standards, including estimated completeness at 97.2%, contamination at 1.87%, and the presence of 16 S rRNA genes (1529 bp) and 40 tRNAs.

**Description of *Atabeyarchaeum* gen. nov.** (A.ta.bey.ar.chae'um N.L. fem. n. Atabey, the Mother Goddess in Taíno mythology; N.L. neut. n. archaeum, ancient, an archaeon from Gr. adj. archaios; N.L. neut. n. Atabeyarchaeum, an archaeon of Atabey, the Mother Goddess)

The type species is *Atabeyarchaeum deiterrae* from the complete genome "Atabeyarchaeia-1" submitted by this study. Seven additional representatives are released by this study, including another complete genome "Atabeyarchaeia-2".

**Description of *Atabeyarchaeaceae* fam. nov.** A.ta.bey.ar.chae.a.ce'ae N.L. neut. n. *Atabeyarchaeum*, the type genus of the family; -aceae, ending to denote a family; N.L. fem. pl. n. *Atabeyarchaeaceae*, the *Atabeyarchaeum* family.

**Description of *Atabeyarchaeales* ord. nov.** A.ta.bey.ar.chae'a.les N.L. neut. n. *Atabeyarchaeum*, the type genus of the order; -ales, ending to denote an order; N.L. fem. pl. n. *Atabeyarchaeales*, the *Atabeyarchaeum* order.

**Description of Atabeyarchaeia class. nov.** A.ta.bey.ar.chae'ia N.L. neut. n. *Atabeyarchaeum*, the type genus of the class; -ia, ending to denote a class; N.L. neut. pl. n. Atabeyarchaeia, the *Atabeyarchaeum* class.

**Description of Frey- taxonomy.** To support the previously published lineage named 'Ca. Freyarchaeota', we also submit the complete "Freyarchaeia" genome as type material for the class 'Ca. Freyarchaeia', order 'Ca. Freyarchaeales', family 'Ca. *Freyarchaeaceae*', genus 'Ca. *Freyarchaeum*', and species 'Ca. *Freyarchaeum deiteterre*, which have been registered via SeqCode (seqco.de/r:igzul8qd).

Description of *Freyarchaeum deiteterre* sp. nov. (de.i.ter'rae L. fem. n. *dea*, deity; L. fem. n. *terra*, Earth; N.L. gen. n. *deiterrae*, of an Earth deity). A soil-associated uncultured lineage, represented by the complete genome "Freyarchaeia" (3.58 Mbp), NCBI BioSample SAMN40176693 in BioProject PRJNA1050611, recovered from California soil sediments. According to SeqCode Registry v1.0.3, the submitted complete genome passes data quality standards, including estimated completeness at 98.13%, contamination at 4.21%, and the presence of 16 S rRNA genes (1264 bp) and 38 tRNAs.

**Description of *Freyarchaeum* gen. nov.** Frey, named after the Norse Goddess most commonly associated with love and fertility; archaeum [from Gr. adj. archaios, -e, -on] ancient, archaeon. Growth condition characteristics. The type species is *Freyarchaeum deiteterre* from the complete genome "Freyarchaeia" submitted by this study.

**Description of *Freyarchaeaceae* fam. nov.** *Freyarchaeaceae* (Frey.ar.chae.a.ce'ae), named after the type genus *Freyarchaeum* gen. nov. The description is the same as for *Freyarchaeum* gen. nov. with the following additions; -aceae ending to denote a family.

**Description of *Freyarchaeales* ord. nov.** *Freyarchaeales* (Frey.ar.chae'a.les), named after the type genus *Freyarchaeum* gen. nov. The description is the same as for *Freyarchaeum* gen. nov. with the following additions; -ales ending to denote an order.

**Description of Freyarchaeia class. nov.** Freyarchaeia (Frey.ar.chae'ia), named after the type genus *Freyarchaeum* gen. nov. The description is the same as for *Freyarchaeum* gen. nov. with the following additions; -ia ending to denote a class.

**Reporting summary**
Further information on research design is available in the Nature Portfolio Reporting Summary linked to this article.

## Data availability
The metagenome-assembled genomes reported in this study can be accessed via the ggKbase database (https://ggkbase.berkeley.edu/SRVP_asgard/organisms). The data used in this study, including all metagenomic and metatranscriptomic reads associated with the reported genomes, have been deposited in the NCBI database under BioProject ID PRJNA1050611, SRA: SRX24933389, SRX24933388, SRX24933387, SRX24933386, SRX23813130, SRX23813131, and SRX23813132. Protein sequences used to generate the alignments and phylogenetic trees, as well as AlphaFold predicted structures, are available via Zenodo (https://doi.org/10.5281/zenodo.11671770). Additional supporting analyses are provided in the Supplementary Information/Source Data file.

## Code availability
The software used in this manuscript is publicly available. No new code was generated.

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

## Acknowledgements

We thank Basem Al-Shayeb for his contribution to field work and generation of sequence datasets, and Shufei Lei and Jordan Hoff for bioinformatics support. We thank Dr. Pok Leung for the discussions on archaeal hydrogenase classification. We also thank Dr. Adam Panagiotis for the helpful discussion about methyltransferases from methanogens and Tristan Wagner for the helpful discussion. We are grateful to Dr. Luke Oltrogge and Dr. Daniel Gittins for their discussions about the multimeric structures modeled using AlphaFold. We thank M. Palmer for guidance with SeqCode nomenclature registration. This publication is based on research in part funded by the Bill & Melinda Gates Foundation (Grant Number: INV-037174 to J.F.B.). The findings and conclusions contained within are those of the authors and do not necessarily reflect positions or policies of the Bill & Melinda Gates Foundation. University of California Dissertation-Year Fellowship (to L.E.V.A.). Stengl-Wyer Graduate Fellowship and University of Texas at Austin Graduate Continuing Fellowship (to K.E.A.). We thank the Innovative Genomics Institute for financial support. Moore-Simons Project on the Origin of the Eukaryotic Cell, Simons Foundation grant 73592LPI (B.J.B.) and Simons Foundation early career award 687165 (to B.J.B.), and Simon Foundation (Award LI-SIAME-00002001) (to B.J.B.). This work was supported by the U. S. Department of Energy Award Number DE-SC0016240 (to D.F.S).

## Author contributions

The study was designed by L.E.V.A. and J.F.B. Samples collection and nucleic acid extractions were performed by L.E.V.A., M.C.S., J.F.B., A.C.C., and J.W.R. Metagenomic data was generated by L.E.V.A., M.C.S., R.S., A.C.C., J.W.R., and J.F.B. Genome binning was done by J.F.B., L.E.V.A., M.C.S., A.C.C., and J.W.R. Complete Asgard genome curation was conducted by J.F.B. and L.E.V.A. Phylogenetic analyses were conducted by K.E.A. and L.E.V.A. Metabolic annotation and analysis done by K.E.A., L.E.V.A. and V.D.A. Metatranscriptome analyses were performed by L.E.V.A. with input from A.C.C. and M.C.S. provided knowledge on metabolism of archaea. C.G. provided knowledge on hydrogenases and archaeal metabolism. V.K. and L.E.V.A. performed tRNAs and selenoproteins analysis. L.L. generated and analyzed the nanopore datasets. D.F.S. and B.J.B. provided feedback on the study design and methodology. J.F.B., D.F.S., and B.J.B. provided resources and funding. K.E.A. and L.E.V.A. registered the genomes via SeqCode. L.E.V.A. and J.F.B. wrote the manuscript, with significant contributions by K.E.A., V.D.A., and M.C.S., and input from all authors. All authors read and approved the manuscript.

## Competing interests

J.F.B. is a co-founder of Metagenomi. D.F.S. is a co-founder and scientific advisory board member of Scribe Therapeutics. L.L. is an employee of Oxford Nanopore Technologies, Inc., and is a stock or stock option holder of Oxford Nanopore Technologies plc. The other authors declare that they have no competing interests.
