## [Peer Review File · Nature Communications]

REVIEWER COMMENTS

Reviewer #1 (Remarks to the Author):

Overall, I think the authors have addressed my concerns. Some overhead statements in previous version have been changed. So I have no more comments.

Reviewer #2 (Remarks to the Author):

I previously reviewed an earlier version of this manuscript for Nature Microbiology, and it seems the authors had the manuscript transferred to Nature Communications after addressing my previous comments. In general, I think the authors did a good job addressing my major concerns and I think that with these changes their conclusions are much more supported by the data presented. However, there are still a couple of comments I have, and some suggestions for improvement (in particular see the need for a potential revision of the legend in Figure 1). I think with these minor changes the manuscript should be suitable for publication.

Specific points and comments:

The reviewers write in their rebuttal that "We concur that there are some parallels with marine sediment archaea. In fact, this is an interesting point - and it might not have been this way. Only by analyzing the genomes can we identify commonalities and differences. These are the first genomes recovered from a soil ecosystem - a significant contribution to our understanding of the diversity and distribution of Asgard archaea. Thus, we do not agree that this is a "weakness"."

I agree with the authors that this is an interesting point (which is why I made it). Since we are in agreement on this, I think it would be a good idea for the authors to add a bit more discussion to the manuscript in this regard. Namely, how about a new paragraph in the discussion about how acetogenesis or homoacetogenesis might be a common metabolism between marine and wetland soil Asgard.

Regarding their responses to the lack of replication of metatranscriptomes, I appreciate the challenges associated with this - particularly for challenging subsurface samples. I think that even without the replicates the data are useful, but need to be appropriately presented. This was a crux of my original review. I think this is now much better in the revised version. However, I recommend to change the wording on line 899 from "were effective in minimizing technical variation" to "should help minimize technical variation" (or something similar). It is also appreciated that the authors have removed, or contextualized, the phrases "highly transcribed" and "weakly transcribed", given the lack of replication.

Regarding the comment in the rebuttal that "It isn't clear to us why a single, properly prepared RNA extract would give very different results. Of course, there will be differences due to RNA preparation methods (also true of DNA, protein, metabolite preparations, and for both pure cultures and real samples) and variation due to the spatial inhomogeneity that is intrinsic to soil. "

I would like to point out, that if you extract RNA from soil or sediment from a single sample, and then make three metatranscriptome libraries from the same RNA extract, you will very likely see that these three libraries contain a large amount of variation between them. I know this from personal experience but I can't prove this is the same for every sample type (try it out sometime, you might be surprised at the variation you get from the same tube of RNA!). It probably has something to do with low levels of RNA extracted and the fact that RNA hydrolyzes extremely quickly. Nevertheless, I think the authors did a good job rephrasing the text and conclusions to adjust for what their data allows for. Even without the replicates it's still helpful to see what genes are and are not expressed between the three genomes they found.

The authors do a good job addressing the comments about potential lack or presence of methanogenesis by the analysis in Figure S1. The high expression of MCR from methanogens is convincing to me, and the text on lines 85-86 is appreciated.

Figure 1 legend: It is stated that Archaea dominate deep regions of wetland soil, but I don't see any data in this figure indicating that Archaea dominate the microbial community. Please show some

data indicating that archaea are more abundant than bacteria in this sample, or remove or rephrase this sentence. I didn't see any other mention of this in the main text, and could not find any comparison of archaeal abundance to bacterial abundance so I think the sentence needs to be removed/rephrased.

Response to referees' comments:

Reviewer #1 (Remarks to the Author):

Overall, I think the authors have addressed my concerns. Some overhead statements in previous version have been changed. So I have no more comments.

We thank the reviewer for all their helpful comments.

Reviewer #2 (Remarks to the Author):

I previously reviewed an earlier version of this manuscript for Nature Microbiology, and it seems the authors had the manuscript transferred to Nature Communications after addressing my previous comments. In general, I think the authors did a good job addressing my major concerns and I think that with these changes their conclusions are much more supported by the data presented.

Thank you for your thorough review of our revised manuscript. We are pleased to hear that our changes have addressed your major concerns from the previous submission to Nature Microbiology. Your feedback has been valuable in strengthening the overall quality of the manuscript.

However, there are still a couple of comments I have, and some suggestions for improvement (in particular see the need for a potential revision of the legend in Figure 1). I think with these minor changes the manuscript should be suitable for publication.

We have carefully considered the remaining comments and suggestions. In particular, we have revised the legend for Figure 1 to accurately reflect the sampling location and provide a more detailed description of the figure panels. The updated figure title now reads:

"Figure 1 Asgard archaea discovered in wetland soil from Lake County, California, USA."

Specific points and comments:

The reviewers write in their rebuttal that "We concur that there are some parallels with marine sediment archaea. In fact, this is an interesting point - and it might not have been this way. Only by analyzing the genomes can we identify commonalities and differences. These are the first genomes recovered from a soil ecosystem - a significant contribution to our understanding of the diversity and distribution of Asgard archaea. Thus, we do not agree that this is a "weakness"."

I agree with the authors that this is an interesting point (which is why I made it). Since we are in agreement on this, I think it would be a good idea for the authors to add a bit more discussion to the manuscript in this regard. Namely, how about a new paragraph in the discussion about how acetogenesis or homoacetogenesis might be a common metabolism between marine and wetland soil Asgard.

We appreciate your agreement that the parallels between marine sediment and wetland soil Asgard archaea are an interesting discovery in our analysis. We concur that discussing the potential commonalities in acetogenesis or homoacetogenesis between these two environments would strengthen our manuscript.

As per your suggestion, we have added the following paragraph to the Discussion section:

Lines 402-415: “Our results suggest acetogenesis is a common metabolic feature shared by marine sediment and wetland soil-associated Asgardarchaeota. The soil metagenomes described here showed potential for acetogenesis and homo-acetogenesis through WLP. Supporting previous analysis indicates acetogenesis is an ancestral trait within this phylum⁴. Homoacetogenic pathways have also been described in several marine Asgard lineages, including Lokiarchaeia, Sifarchaeia, Thorarchaeia, and Wukongarchaeia^{3,5,6,27,11}. We were able to identify both acetogenesis and homo-acetogenesis genes in the two lake sediment Atabayarchaeia MAGs and Freyarchaeia MAGs extracted from deep-sea (Guaymas Basin) and hot spring sediments (Tibet Plateau, Yunnan Province, and Radiata Pool)^{4,12}. The ability to fix CO₂ and produce acetate via the WLP could provide a competitive advantage for Asgard archaea in marine and terrestrial environments, where competition may limit labile organic carbon sources. Acetate production in today’s Asgard archaea from various environments may support the growth of other microbial community members, including acetoclastic methanogens.”

Regarding their responses to the lack of replication of metatranscriptomes, I appreciate the challenges associated with this - particularly for challenging subsurface samples. I think that even without the replicates the data are useful, but need to be appropriately presented. This was a crux of my original review. I think this is now much better in the revised version. However, I recommend to change the wording on line 899 from "were effective in minimizing technical variation" to "should help minimize technical variation" (or something similar). It is also appreciated that the authors have removed, or contextualized, the phrases "highly transcribed" and "weakly transcribed", given the lack of replication.

We thank reviewer #2 for understanding and suggesting the revised wording on line 899. Line 899 now reads: “Despite the absence of technical replicates, these preparatory measures should help minimize technical variation.”

Regarding the comment in the rebuttal that "It isn't clear to us why a single, properly prepared RNA extract would give very different results. Of course, there will be differences due to RNA preparation methods (also true of DNA, protein, metabolite preparations, and for both pure cultures and real samples) and variation due to the spatial inhomogeneity that is intrinsic to soil. "

I would like to point out, that if you extract RNA from soil or sediment from a single sample, and then make three metatranscriptome libraries from the same RNA extract, you will very likely see that these three libraries contain a large amount of variation between them. I know this from personal experience but I can't prove this is the same for every sample type (try it out sometime, you might be surprised at the variation you get from the same tube of RNA!). It probably has something to do with low levels of RNA

extracted and the fact that RNA hydrolyzes extremely quickly. Nevertheless, I think the authors did a good job rephrasing the text and conclusions to adjust for what their data allows for. Even without the replicates its still helpful to see what genes are and are not expressed between the three genomes they found.

Thank you for sharing your insights. We are pleased that you find our rephrasing appropriate.

The authors do a good job addressing the comments about potential lack or presence of methanogenesis by the analysis in Figure S1. The high expression of MCR from methanogens is convincing to me, and the text on lines 85-86 is appreciated.

We thank the reviewer for their positive feedback.

Figure 1 legend: It is stated that Archaea dominate deep regions of wetland soil, but I don't see any data in this figure indicating that Archaea dominate the microbial community. Please show some data indicating that archaea are more abundant than bacteria in this sample, or remove or rephrase this sentence. I didn't see any other mention of this in the main text, and could not find any comparison of archaeal abundance to bacterial abundance so I think the sentence needs to be removed/rephrased.

We have modified the figure as suggested and removed this sentence from the figure legend, which now reads:

"Figure 1 "Asgard archaea discovered in wetland soil from Lake County, California, USA."